# Food Insecurity and Water Insecurity in Rural Zimbabwe: Development of Multidimensional Household Measures

**DOI:** 10.3390/ijerph18116020

**Published:** 2021-06-03

**Authors:** Nadia Koyratty, Andrew D. Jones, Roseanne Schuster, Katarzyna Kordas, Chin-Shang Li, Mduduzi N. N. Mbuya, Godfred O. Boateng, Robert Ntozini, Bernard Chasekwa, Jean H. Humphrey, Laura E. Smith

**Affiliations:** 1Department of Epidemiology and Environmental Health, The State University of New York, University at Buffalo, Buffalo, NY 14214, USA; bibinadi@buffalo.edu (N.K.); kkordas@buffalo.edu (K.K.); 2Department of Nutritional Sciences, School of Public Health, University of Michigan, Ann Arbor, MI 48109, USA; jonesand@umich.edu; 3Center for Global Health, School of Human Evolution and Social Change, Arizona State University, Tempe, AZ 85281, USA; roseanne.schuster@asu.edu; 4School of Nursing, The State University of New York, University at Buffalo, Buffalo, NY 14214, USA; chinshan@buffalo.edu; 5Global Alliance for Improved Nutrition, Washington, DC 20036, USA; mmbuya@gainhealth.org; 6Department of Kinesiology, College of Nursing and Health Innovations, The University of Texas at Arlington, Arlington, TX 76019, USA; godfred.boateng@uta.edu; 7Zvitambo Institute for Maternal and Child Health Research, Harare, Zimbabwe; r.ntozini@zvitambo.com (R.N.); b.chasekwa@zvitambo.com (B.C.); jhumphr2@jhu.edu (J.H.H.); 8Department of International Health, Johns Hopkins Bloomberg School of Public Health, Baltimore, MD 21205, USA; 9Department of Population Medicine and Diagnostics, Cornell University, Ithaca, NY 14853, USA

**Keywords:** food insecurity, water insecurity, households, dimensions, measures

## Abstract

*Background*: With millions of people experiencing malnutrition and inadequate water access, FI and WI remain topics of vital importance to global health. Existing unidimensional FI and WI metrics do not all capture similar multidimensional aspects, thus restricting our ability to assess and address food- and water-related issues. *Methods*: Using the Sanitation, Hygiene and Infant Nutrition Efficacy (SHINE) trial data, our study conceptualizes household FI (*N* = 3551) and WI (*N* = 3311) separately in a way that captures their key dimensions. We developed measures of FI and WI for rural Zimbabwean households based on multiple correspondence analysis (MCA) for categorical data. *Results*: Three FI dimensions were retained: ‘poor food access’, ‘household shocks’ and ‘low food quality and availability’, as were three WI dimensions: ‘poor water access’, ‘poor water quality’, and ‘low water reliability’. Internal validity of the multidimensional models was assessed using confirmatory factor analysis (CFA) with test samples at baseline and 18 months. The dimension scores were associated with a group of exogenous variables (SES, HIV-status, season, depression, perceived health, food aid, water collection), additionally indicating predictive, convergent and discriminant validities. *Conclusions*: FI and WI dimensions are sufficiently distinct to be characterized via separate indicators. These indicators are critical for identifying specific problematic insecurity aspects and for finding new targets to improve health and nutrition interventions.

## 1. Introduction

Globally, over two billion people do not have regular access to safe, nutritious and sufficient food [1], while about four billion are exposed to water stress at least once a month [2]. However, the definitions of both food insecurity (FI) and water insecurity (WI) go beyond only inadequate access. The most widely accepted definition of food security is “when all people at all times have physical, social and economic access to sufficient, safe ad nutritious food to meet their dietary needs and food preferences for an active and healthy life” [3]. Water security is analogous to food security and refers to “safe and reliable access to adequate quantity and quality of water for consumption, economic production and cleanliness” [4,5]. Both these definitions identify multiple dimensions of FI and WI, like availability, access, quality and safety, and reliability of supply [3,6,7]. Nevertheless, inconsistencies exist between these internationally recognized definitions and the ways in which the concepts of FI and WI are applied in research and policy [8,9].

Decades of work on FI have produced a diverse range of metrics at the household-level [8]. Many of these existing metrics are often used interchangeably even though they capture different combinations of FI dimensions [8,10]. For example, the commonly used Household Food Insecurity Access Scale (HFIAS) [11] and Food Insecurity Experience Scale (FIES) [12] capture economic food access and food sufficiency, the Household Dietary Diversity Score (HDDS) captures diet quality [13] and the Coping Strategy Index (CSI) captures food access experiences in emergency settings [14]. These metrics, due to their respective conceptualization and intended use, are correlated, but not equivalent [10,15,16,17]. In empirical comparisons of some of these household measures, different estimates of FI prevalence were reported for the same population at the same time point [16]. For example, CSI in Ethiopia identified 58% food insecure households, but HFIAS detected 66% [16]; in Bangladesh and India, 34% of households with adequate caloric intake were classified as food insecure based on CSI or HFIAS [17]. As a result of these challenges, the Food and Agriculture Organization (FAO) recommends the development and application of multiple indicators to distinguish FI dimensions, to improve accuracy and specificity of FI assessments, and to prevent misclassification based on dimensions of FI [18].

Assessment of household-level WI is less established than FI, and few cross-culturally valid metrics exist. The most common measures of WI developed by the World Health Organization (WHO) categorize water access based on fetching time, and water sufficiency based on quantity of water available per person [19]. Another measure from the Joint Monitoring Programme (JMP) for Water Supply and Sanitation approach distinguishes “improved” water access based on the type of the main water source [20]. Population- and context-specific WI scales have also been developed in Kenya [21,22], rural Ethiopia [23,24], urban Nepal (Household Water Insecurity Scale (HWIS)) [25], slums of India (Water Insecurity Experience Scale (WIES)) [26], Uganda (Household Water Insecurity Access Scale (HWIAS) [27], Bolivia [28], Brazil [29], Jamaica (Water Accessibility Index (WAI)) [30], and the colonias along the US-Mexico border [31]. These metrics vary in the WI dimensions they attempt to capture. For instance, the Household Water Insecurity Experiences (HWISE) Scale, a unidimensional 12-item cross-cultural scale, captures emotions and behaviors predominantly in response to inadequate water supply [5]. The WAI captures only water access using seven components including affordability, source and collection time [30]. Although these existing WI measures are instrumental in identifying water insecure households, further distinguishing between WI dimensions will allow targeted interventions and policy decisions.

FI and WI have been shown to contribute directly and indirectly to undernutrition [1,7], psycho-social stress [25,28,32,33,34], and increased risk of infectious and chronic diseases [1,7]. They often chronically co-exist within the same households [35]. The high prevalence and co-occurrence of FI and WI may have synergistic effects on adverse health outcomes [36]. This may have significant implications for achieving sustainable development goals (SDG) two, three and six to “end hunger and all forms of malnutrition”, “ensure healthy lives and promote well-being for all” and “ensure availability and sustainable management of water and sanitation for all” [37], respectively. However, few studies have been able to explore FI and WI concurrently as environmental stressors in causal pathways to specific health and nutrition outcomes [36,38,39,40,41]. This is in part because of the complexity surrounding the conceptualization and the measurement challenges of FI and WI as described above [8,42,43].

More than one third of the world’s food insecure and water insecure people live in Sub-Saharan Africa (SSA) [44,45]. In Zimbabwe, economic crises, recurrent droughts, and depletion of ground water are causing severe food and water shortages [46]. Approximately 30% of the rural Zimbabwean population is undernourished [47] and obtains water from unprotected sources [48]. A prior study in rural Zimbabwe demonstrated the value of using multiple FI indicators in designing and evaluating interventions, when CSI and HDDS gave different prevalence of food insecure households [49]. The same is likely true for WI, where the application of distinct indicators for each dimension will be more valuable. Our objectives were therefore to develop separate multidimensional measures for household FI and WI, and to test their internal, predictive, convergent and discriminant validities in the context of rural Zimbabwe.

## 2. Materials and Methods

### 2.1. Study Setting and Population

Data for the development of the household measures of FI and WI were obtained from the Sanitation, Hygiene and Infant Nutrition Efficacy (SHINE) trial. The trial’s primary objectives were to test the independent and combined effects of an improved water, sanitation and hygiene (WASH) intervention, and an improved infant and young child complementary feeding (IYCF) intervention on stunting and anemia among rural Zimbabwean children. The design, protocol, and primary outcomes have been published elsewhere [50,51,52]. Briefly, SHINE was a four-arm cluster-randomized community-based 2 × 2 factorial trial conducted in two rural districts in Zimbabwe: Shurugwi and Chirumanzu. The two districts were divided into 212 clusters which were then randomly allocated to one of the four trial arms: (1) Standard of Care (SOC), (2) SOC + IYCF, (3) SOC + WASH and (4) IYCF + WASH. Recruitment occurred between 22 November 2012 and 27 March 2015. Village health workers (VHWs) employed by the Zimbabwe Ministry of Health and Child Care prospectively identified and referred eligible women for the trial. Only women residing permanently in a cluster and who were pregnant at the time of recruitment were enrolled. Written informed consent, in the language of their choice (English, Ndebele, or Shona), was obtained prior to data collection. SHINE was approved by the Medical Research Council of Zimbabwe and the Johns Hopkins University Bloomberg School of Public Health Institutional Review Board.

### 2.2. Data Collection

SHINE included an extensive structured questionnaire to collect detailed information on household, maternal and child characteristics. Baseline data collection spanned the recruitment period mentioned above. A few weeks after obtaining consent, research nurses made home visits for face-to-face interviews with the women. Additional home visits for subsequent data collection were also made at one, three, six, 12- and 18-months post-partum, until the end of the study in July 2017. The questionnaire and data collection protocol are available on OSF at https://osf.io/w93hy/ (accessed on 17 May 2021).

### 2.3. Sample Selection

From the 5280 pregnant women who were recruited, 4675 took part in the baseline interview. For the following analysis, the sample was restricted to households with complete information on the selected food (*N* = 3551) and water (*N* = 3311) variables. Figure 1 illustrates participant inclusion.

The creation of FI and WI measures were carried out in a stepwise manner, starting with item variable selection for inclusion in the quantitative analyses. The next steps included descriptive analyses, item reduction, multiple correspondence analysis (MCA) with extraction and rotation of dimensions, and validity assessments.

### 2.4. Item Variable Selection

The starting point for item selection was the internationally accepted definitions and dimensions of FI [3] and WI [7]. The FAO [3] and Action Contre la Faim (ACF) [53] provide some recommendations for indicators of FI dimensions, while WaterAid [7], Global Water Partnership (GWP) [54] and JMP [20] suggest items for WI dimensions. Indicators relevant to rural Zimbabwe and available from SHINE were then selected. Table 1 provides detailed descriptions of all variables selected to represent each FI and WI dimension. Brief justifications are also provided below for the choice of item variables:

*A.1. Food availability* refers to the food supply aspect of food security [3]. This dimension considers whether food is actually present for the population [55]. At the national-level, this has historically been addressed via the use of food balance sheets of food production and imports. At the rural household-level, food availability may be captured by considering food stocks, presence of markets and ability to produce food. We used three variables to operationalize this dimension: (1) number of days of staple food stocks available for household members to eat according to their needs, (2) availability of a garden where the household grows fruits and vegetables, and (3) the availability of left-over food from the last cooking occasion.

*A.2. Food access* concerns economic, physical and social resources that enable acquisition of sufficient, nutritious and preferred foods in a dignified manner [3]. Physical food access is linked to infrastructure and at the household-level can be captured by considering time spent, distance travelled and transportation to safe food sources. Economic access depends on the ability of households to purchase or barter resources to obtain food [55]. Social access concerns food preferences in terms of taste, health requirements and religious restrictions. It also implies that food is obtained in socially acceptable ways. The following seven household-level variables were considered for this dimension: (1) access to preferred food, (2) food sufficiency for all household members, (3) help required from family and/or friends to obtain food, (4) purchasing or borrowing food on credit, (5) selling assets for food, (6) time from home to food market, and (7) method of transportation to food market.

*A.3. Food utilization* reflects differences in the intra-household allocation of food, nutritional quality of food, and food safety in terms of preparation, handling, and storage [8,55]. Within SHINE, four variables were available as proxies for food utilization: (1) household dietary diversity, (2) handwashing behavior prior to handling food, (3) whether food containers were covered, and (4) food storage location. No information was available as proxy for intra-household allocation, which also depends on age, work load, and other factors.

*A.4. Food stability* covers the barriers and promotors of food security dimensions [8,55]. At the household-level, this can be captured by considering exposures to risks, shocks or vulnerabilities that influence the ability of household to consistently acquire food [55]. The variables most appropriate to represent this dimension from SHINE were household experiences of social, economic, agriculture and health shocks.

*B.1. Water availability* depends on the physical presence of water resources or infrastructure that makes it available in sufficient quantity to households [56]. Sufficient quantities of water must be available for drinking to prevent dehydration (≥5 L per person/day) and for cooking, bathing, hygiene and sanitation (>100 L per person per day) [19]. Within SHINE, two variables were considered: (1) volume of water, calculated from storage capacity of water containers and water collection frequencies, and (2) whether the households had access to water for irrigation purposes.

*B.2. Water access* refers to physical delivery and economic access to water. Methods for assessing water access include the distance to water points, fetching time, and water expenditures [19,57]. Water access is inadequate if households have to travel >1 km or >30 min (return journey) to collect water [19,58]. Water is affordable if households spend <3–5% of their total income on it [59]. Five variables were considered to assess water access: (1) whether the household purchases water, (2) drinking water collection time, (3) distance to drinking water point, (4) non-drinking water fetching time, and (5) distance to non-drinking water point.

*B.3. Water utilization* is meant to reflect the quality and safety of water for drinking and other purposes. Physical quality can be measured by considering the color, smell and taste of the water. Chemical quality and microbiological safety are determined by testing turbidity, total dissolved solids, chlorine levels and the presence of bacterial coliforms in the water. In low-income settings, types of water sources are used as proxy for water quality and safety [19]. For instance, protected sources such as piped water, boreholes and wells are considered microbiologically and chemically safer compared to surface water from rivers or streams. To capture this dimension, three SHINE variables were used: (1) reported satisfaction with the water smell, color and taste, (2) water source for drinking, and (3) water source for non-drinking purposes.

*B.4. Water reliability* refers to whether water supply is consistent or intermittent. Whether water is piped into dwellings or available off premises, it may be periodically or seasonally inaccessible [2]. To assess the reliability of water supply among SHINE households, two variables were considered: (1) whether drinking source and (2) non-drinking source ran dry over the past year.

### 2.5. Statistical Analyses

Separate multiple correspondence analysis (MCA) were conducted on the selected item variables to develop FI and WI measures. MCA for categorical variables is equivalent to exploratory factor analysis (EFA) or principal component analysis (PCA) designed for continuous variables [60]. Analyses were conducted as explained below using Stata Version 16 (StataCorp LLP, College Station, TX, USA) for descriptives, ‘FactoMineR’ [61] and ‘PCAmix’ [62] packages from the software R Version 4.0.2 for MCA and factor rotation, and MPlus Version 8.4 (Muthén & Muthén, Los Angeles, CA, USA) for validity tests.

#### 2.5.1. Descriptives

First, we looked at the distributions of participants across the categories of each item variable using frequencies and percentages. Variables with categories reporting frequencies of ≤5% or ≥95% were excluded.

#### 2.5.2. Item Reduction

Second, we ran polychoric correlations on all variables. Items indicating negative correlations and those without adequate variance (<0.1) were dropped. We also used the Kaiser-Meyer-Olkin (KMO) measure for sampling adequacy and Barlett’s test of sphericity to ensure robustness of our approach. We then carried out MCA on the remaining variables. Scree plots were used to decide the number of dimensions for extraction. We investigated factor extraction using oblique (geomin) and orthogonal (varimax) rotations. Since correlations among the extracted factors were small (<0.5), we report varimax-rotated loadings in our results. Dimensions extracted were interpreted and named based on the variables that loaded on them from the theoretical framework (Table 1). We report the squared correlation ratios between each item variable and dimension, eigenvalues and percentage explained variances. Squared correlation ratios <0.20 were not considered relevant in explaining a dimension. We then used post-estimation commands in R to obtain standardized dimension scores for individual households.

#### 2.5.3. Validity Assessments

Validity refers to the extent to which certain measures are acceptable indicators for what they are intended to capture [63]. We tested four types of validity for our FI and WI measures: internal, predictive, convergent and discriminant. These are briefly described in Table 2 with an explanation of their purpose and statistical methods used. For internal validity, we assessed multidimensional model fit via confirmatory factor analysis (CFA) in two groups: (1) a sub-sample of the baseline participants constituting 60% of the dataset, and (2) the same baseline households more than 18 months after the baseline interview. We used model fit statistics such as root mean square error of approximation (RMSEA), standardized root mean square residual (SRMR), comparative fit index (CFI), and Tucker Lewis index (TLI). Satisfactory fit was determined using recommended arbitrary cut-offs of RMSEA ≤ 0.05, SRMR ≤ 0.08, CFA ≥ 0.95 and TLI ≥ 0.95 [64,65]. CFA was performed in MPlus using geomin rotation with diagonally weighted least squares estimator (WLSMV).

For predictive, discriminant and convergent validity, we used a group of exogenous variables, also obtained from the SHINE trial. Self-reported perceived health status of women was measured using an adapted version of the RAND Health Survey [66]. Scores for perceived health status ranged from 0 to 5 units, with 0 indicating least healthy and 5 most healthy [67]. The Zimbabwe-validated version of the 10-question Edinburgh Postnatal Depression Scale (EPDS) was used to assess depression among the women [68]; those with a score ≥12 out of 30 were classified as clinically depressed. Household receiving food aid over the past 12 months from government or other organizations (yes/no) was self-reported by women. Usual frequency of water collection was reported as daily, weekly or monthly. HIV-status of the participating women was determined via rapid blood tests performed by trained nurses [50]. Household socio-economic status (SES) was based on a household wealth index [69]. Seasonality was determined based on the date of interview; hungry season was from January through March and rainy season was from November through March. These variables were used as predictors in simple regressions to estimate associations with FI and WI dimension scores from MCA.

#### 2.5.4. Sensitivity Analysis

We tested the robustness of the MCA results after accounting for missingness in the selected items. Almost all variables had <10% missing values (Appendix A). Lower SES, HIV-status and interview months were found to influence missingness (Appendix A). To account for missing data uncertainty, we imputed missing variables using the multiple imputation by chained equations (MICE) method via the ‘MICE’ function from the ‘missMDA’ package in R [70]. We then re-ran MCA by including the additional households with imputed variables. Only households with less than three imputed variables were used for sensitivity analysis. The sample size increased considerably (*N* = 4622 for FI and *N* = 4575 for WI).

## 3. Results

### 3.1. Participant Characteristics

Table 3 summarizes the distribution of households and participating women according to socio-demographic and food- and water-related characteristics. The two samples used to generate FI and WI measures were similar for all variables. The average age of the respondents was 26 ± 7 years. Approximately the same proportion of participants were randomized in the four SHINE trial arms, 15% were living with HIV, 44% were interviewed during the rainy season and 28% during the hungry season, more than 40% of the participants had completed secondary school, more than half had children, and 45% were of Apostolic faith. At least 6% of women were clinically depressed. More than 90% were partnered and were not generating income outside the home.

### 3.2. Multiple Correspondence Analysis (MCA)

Descriptive and item reduction analyses suggested the removal of three food variables (transportation, leftover food and food container) and two water variables (water purchase and irrigation water) from the initial list (Table 1). The KMO statistic was above 0.74 for food and 0.48 for water. Barlett’s sphericity tests were significant for both food and water samples (*p* < 0.01). These two indicators suggest that the final 15-item food dataset is adequate to further explore underlying latent constructs. Adequacy of the 12-item water dataset is poor according to the KMO statistic. However, Barlett’s sphericity statistic suggests substantial inter-item correlation between the water variables, indicating that factor analysis may still be useful.

A visualization of the scree plots for both food and water MCA showed that the eigenvalues plateaued after the third dimension for FI (Figure 2). For WI, there appears to be multiple inflection points, at dimensions three and five. However, we consider only the first three dimensions, since beyond this, the water items were cross-loaded at lower factor loadings.

We therefore chose three dimensions for FI. They are named as follows based on the items that loaded on them (Table 4): (1) “**poor food access**” (food not preferred, insufficient food, food help and food on credit), (2) “**household shocks**” (economic shocks, agriculture shocks and health shocks), and (3) “**low food availability and quality**” (stock of staple food, garden and household diet diversity). These dimensions accounted for a cumulative variance of 20.12%. Similarly, the WI dimensions are named according to the characteristics they capture (Table 4): (1) “**poor water access**” (time to drinking source, distance to drinking source, time to non-drinking source, distance to non-drinking source), (2) “**poor water quality**” (drinking source, non-drinking source, water satisfaction), and (3) “**low water reliability**” (whether drinking and non-drinking sources ran dry). Together, these dimensions accounted for a cumulative variance of 31.36%.

The median dimension scores indicated low insecurity in our population with wide interquartile ranges. Nevertheless, the minimum and maximum values ranged from −1.04 to 3.93 units suggesting variations in FI and WI across households. Poor water access was significantly correlated with all FI dimensions; poor water quality was correlated with poor food access and low food quality and availability, and low water reliability was correlated with poor food access and household shocks. However, none of the correlations were very high (r < 0.15 for all pairwise associations) (Appendix A).

### 3.3. Validity Assessments

Table 5 summarizes model fit statistics for internal validity of the multidimensional FI and WI measures. There was strong support for both FI and WI measures, with at least two of the four indices indicating satisfactory cut-offs.

Almost all assessments related to predictive, discriminant and convergent validities were in the expected direction, although all were not statistically significant (Table 6). Higher perceived health status was associated with lower FI and WI scores. Depression, lower SES and living with HIV were associated with higher FI and WI scores. Households interviewed from January through March (hungry season) scored higher on FI dimensions, while those interviewed from April through October (dry season) had poorer water access. Households receiving food aid had higher scores on shocks. Less frequent water collection was associated with poorer water access.

Our sensitivity analyses, after imputation of missing values for the relevant food and water item variables, confirmed the robustness of the results presented. The number of dimensions identified, items loading on each dimension, correlation ratios and eigenvalues were similar to the complete case analysis (Appendix A).

## 4. Discussion

The goal of this study was to develop new measures of FI and WI that are cognizant of their multidimensionality to advance the discussion on impactful nutrition and health interventions for vulnerable populations. We used rigorous analytical procedures to develop these measures among rural Zimbabwean households. Each of the FI and WI measures obtained consist of three dimensions. The multidimensionality observed through the development process is in part consistent with the definitions of both FI [3] and WI [7]. The distinction between the dimensions from our household measures of FI and WI provide additional depth that may complement existing FI and WI metrics. The quantification of the dimensions will advance our understanding of their prevalence and consequences for health and well-being. We named our measures the multidimensional household food insecurity (MHFI) and the multidimensional household water insecurity (MHWI), respectively.

### 4.1. Food Insecurity (FI)

MCA with 15 food-related variables resulted in the identification of multiple dimensions of FI as theorized previously (Table 1). The first dimension refers to **“poor food access”** through quantity, affordability and food preference. Access to food includes the social, physical and economic aspects [3,55]. However, this first dimension captures only the social and economic access to food. The variables ‘time to market’ and ‘mode of transportation to market’, that represent physical access to food, did not load on any dimension. The second FI dimension, **“household shocks”**, describes the reliability component of food supply and includes households’ experiences of economic shocks through loss of employment or assets; agricultural shocks through loss of crops and livestock; and health shocks through death, disease and injury of household members. Social shocks such as conflict or legal problems were not retained in this dimension. The third dimension, **“low food availability and quality”**, includes poor household dietary diversity, low stock of staple food and lack of household garden. This dimension partially encompasses several theoretical components of FI: utilization (dietary diversity) and availability (stock of staple food, having a garden). This may be because after the item reduction step, only two variables were left to represent availability (stock of staple food and having a garden) and two to represent utilization (handwashing and food storage location) in the MCA model.

In contrast to existing FI scales (HFIAS, FIES, CSI) whose internal consistency arise from assessing similar constructs, our FI dimensions reflect different conceptual constructs due to the use of variables with disparate measurement approaches and recall periods (Table 1). These differences may impede our ability to interpret and compare the FI dimensions to each other, and to existing scales. Nevertheless, our three-dimensional measure of FI was found to be valid within a test sample of the SHINE population and across time (Table 5). Moreover, it is possible for households with similar scores on one of the existing FI metrics to have different characteristics on individual FI dimensions [10,43,49]. This is an important limitation for exploring impact pathways or to identify relevant intervention targets, because composite scores do not inform on which aspects of availability, access, utilization or stability to modify. Therefore, the three FI dimensions identified in our population contribute to addressing the need for multiple indicators to improve the identification of food insecure households [3,18,43,71].

### 4.2. Water Insecurity (WI)

In our study, 12 water-related variables loaded on three dimensions (Table 4). The first dimension refers to **“poor water access”** and includes time and distance variables. The second dimension, **“poor water quality”**, includes types of water sources and degree of satisfaction with water quality. Finally, the third dimension refers to **“low water reliability”** and includes information on whether water for drinking and non-drinking purposes was unavailable at any point. Of the four hypothesized dimensions of WI (Table 1), availability in terms of water quantity was the only dimension not identified, likely due to the low number of variables in SHINE to represent this dimension thoroughly. Interestingly, neither Stevenson et al. (2012) in Ethiopia [24] nor Tsai et al. (2016) in Uganda [27] reported any correlations between their experience-based WI scores and water quantity. However, HWISE-4 (shorter version of the HWISE questionnaire) does capture the experience of adequate quantity of drinking water for consumption [72]. Although our WI model was found to be valid at baseline, it was not completely supported at 18 months, suggesting that additional water-related information may be needed to better represent WI over time.

Our study, like some others [5,21,24,27], acknowledges the importance of water for both non-drinking (e.g., laundry, bathing, cooking, irrigation, etc.) and drinking purposes. Recent efforts to develop WI metrics have mostly focused on composite scales that capture at least one, but not all, WI dimensions (HWISE, WIES, HWIAS, WAI) [5,21,26,27,30,31]. Like for FI, using unidimensional composite scales masks the contributions of the multiple WI dimensions. For example, the initial HWISE questionnaire included questions on the taste, smell, color, treatment, reliability, and usability of water, among others. However, the focus was on how people experienced those things and only some aspects were retained in the final scale [5]. This is in contrast to our analyses, where water quality was identified as a distinct dimension. Our water quality dimension has the added advantage of including types of water sources, also considered reliable and factual by the JMP [20]. Similarly, for water access, the WHO uses water fetching time as a simple measure [19], while the HWISE scale was correlated with time to water source [5]. Our water access dimension includes distance to water point, and as such, captures additional access information. Distance travelled and time taken for water collection form part of the WAI [30]. Whereas in our analyses, reliability of water supply was a distinct dimension, it was an integral part of the WAI. This may be explained by the difference in variable measurement: WAI reported time period when water supply is cut-off on any given day while we report any disruption in water supply over the year prior to the interview.

### 4.3. Validity Assessments

Validity of our FI and WI measures was supported in a number of ways (Table 5 and Table 6). Firstly, internal validity was confirmed in two separate groups: a test sample of the baseline population and the SHINE households more than 18 months later. The consistent results produced in both instances indicate structural adequacy of the FI and WI dimensions within this population. Secondly, higher perceived health status was associated with lower FI and WI, while maternal depression was associated with higher FI and WI. These associations, although small, suggest good predictive validity. The results are consistent with findings in similar populations between health conditions and experiential measures of FI [34,73,74,75] and WI [5,21,25,33]. These validity analyses add to the existing literature by considering additional dimensions of FI and WI.

Convergent validity was examined by assessing the relationship between dimension scores with food aid for FI and frequency of water collection for WI. Households having experienced shocks were more likely to receive food aid, which suggests convergent validity. Although the association with poor food access and low quality and availability were not significant, they were in the expected direction. Lower frequency of water collection was associated with poorer water access. Upon further exploration within SHINE, we found that the longer women spent on water collection at any one time, the less often they collected water.

Finally, we evaluated discriminant validity via the associations between FI and WI dimension scores and season of interview, SES and HIV-status. As expected and consistent with international reports, households with lower SES, or those interviewed during the hungry season, and women living with HIV had poorer food access and lower food quality and availability. In the months between the last season’s food stores and current season’s harvesting of crops, food supplies in farming communities run low, and people often use food aid and other coping strategies [53]. Similarly, households at lower SES, especially subsistence farmers, may be less able than those at higher SES to access sufficient and nutritious food [1]. The association between HIV-status and FI is considered bidirectional [76]. People living with HIV have a greater need for adequate food to ensure the success of antiretroviral therapy (ART). At the same time, because they are weakened by their disease, they may be unable to procure food or resources needed to obtain food [77,78]. This may be true of other health outcomes, such as physical health status and depression used for predictive validity. Since we are primarily concerned with associations rather than causation in these validity analyses, although relevant, bidirectionality does not affect our interpretation.

Water access was found to be poorer among households interviewed during the dry season compared to the rainy season. During the dry season, households may need to travel longer distances and spend more time on finding water because their usual water source ran dry, whereas in the rainy season, water sources may be more abundant [27]. SHINE households with low to middle SES had poorer water quality compared to those of high SES, because they were less likely to have access to piped water or protected water sources. The association with HIV status was not significant for any of the WI dimensions. However, a prior study measuring WI on the experiential scale reported that women living with HIV in Kenya were more water insecure [21].

### 4.4. Strengths and Limitations

The strengths of this study make it a worthwhile exploration of new FI and WI measures in an underserved population. First, we were able to use information from over three thousand households to develop and test the structural validity of these new measures. The households are representative of the rural population in Zimbabwe, implying that the FI and WI measures may be valid in other similar rural areas [52]. Second, sensitivity analyses accounting for missing data further increased the sample size and enhanced our confidence in the robustness of the multiple dimensions identified (Appendix A). Third, unlike previous scales, our measures reflect additional theoretical multidimensionality in both FI and WI separately, and allow distinction between key dimensions. We were able to show via the statistical method of MCA that it is possible to come up with and quantify different aspects of FI and WI. We expect that future research will consider using similar methods to distinguish between FI and WI dimensions in other settings. The Gallup World Poll implemented the FIES questionnaire in 2019 for FI monitoring and recently decided to include HWISE for WI. Our method and potentially our MHFI and MHWI measures may serve as supplements for identifying and addressing specific food and water problems in vulnerable populations. Fourth, our study is unique in that it considers the simultaneous development of both FI and WI measures in a rigorous randomized controlled trial like SHINE, which included training of data collectors and quality control that ensured information accuracy. Finally, we recently implemented the questions identified in our study in another survey with a separate group of Zimbabwean households, and found that it took less than 15 min for trained data collectors to obtain the required FI and WI information. For all questions, higher literacy levels shortened the interview time. Since these questions are easy to add and are of low time burden, we hope that this paper further encourages global health scientists and policy makers to think about the individual components that define FI and WI when designing health and nutrition interventions.

Although SHINE’s IYCF intervention showed a reduction in stunting prevalence, the improvement was modest; while SHINE’s WASH intervention showed no improvement [50]. Similar modest improvements have been reported in other IYCF and WASH interventions, in countries such as Bangladesh and Kenya [79,80,81]. It is hypothesized that the impact on stunting reduction could be larger if underlying determinants (e.g., FI and WI) as per the UNICEF’s framework for undernutrition are addressed [82]. In most instances, efforts to mitigate FI and WI are complicated by interactions with social, environmental, and physical processes [83,84]. Therefore, our deconstructed measures of FI and WI provide an opportunity to explore specific aspects of FI and WI on undernutrition and related interventions for better targets. The contribution of our WI measures will be particularly important for the transformative WASH movement, which calls for the radical reduction of fecal contamination in the household environment in LMICs [85,86]. Despite intensive implementation and uptake of low-cost household-based WASH interventions in recent trials [50,80,81], environmental fecal contamination remains pervasive [85]. Our WI measures can be used to assess convenient and adequate access to uncontaminated water, which is central to transformative WASH. This concept is also critical for the utilization dimension of food security, which requires safe water for food handling and preparation.

This study is innovative in its approach for finding multidimensional measures of FI and WI, but it is not without limitations. First, the dimensions explained a small percentage of the cumulative variance between items. This may be a concern, although the percentage variance is generally smaller in MCA than in PCA, because individuals are located in a high K-J dimensional space which gets larger as the number of categories increase [87]. The low variances are not unlike what have been observed in other categorical or binary factor analyses [69,87]. Second, these measures were developed in rural Zimbabwe and among households with pregnant women. Although the methods and overall dimensions are potentially transferable, generalizability outside of this population may not be appropriate. When used in other settings, modifications will be required to the items included for the MCA. For example, water purchase was not relevant in rural Zimbabwe because the majority of its rural population does not buy water. Similarly, water purchase and affordability were not retained in HWISE [5]. However, this is an important access consideration for some populations, like those residing in colonias on the US-Mexico border and in low-income communities in Jamaica [30,88]. For FI, transportation and time to markets may be important among households that do not engage in subsistence farming, e.g., in areas with food scarcity in the USA [89].

Third, the dimensions retained are limited to the variables available from the SHINE questionnaire. We could not capture the experiences of thirst, hunger or emotional distress (e.g., anger, frustration, shame) associated with FI and WI because such information was not collected. Therefore, our household FI and WI measures may be used to complement experience scales such as FIES and HWISE for an all-encompassing view of these resource insecurities. Moreover, to better capture FI utilization, Appendix A suggested by the FAO, like food consumption scores (to determine equitable intra-household food distribution), and utilization of clean utensils for cooking and eating, would have been useful [3]. Similarly, additional information on objective microbiological and physico-chemical assessments of water at point of use, details on intermittent availability of water from all sources, quantification of the amount of water used per household member to determine need-based equitable access, and water sufficiency for purposes other than drinking, would have improved our WI dimensions. We also caution on the ‘poor water reliability’ dimension, created with only two variables, making it prone to quantitative estimation problems [65]. Nevertheless, a two-item dimension is considered reliable if they are highly correlated with each other, as is our case (r = 0.75), but fairly uncorrelated with other variables [90,91].

Finally, another concern may be seasonality, due to the dependence of FI and WI on environmental conditions in a land-locked country like Zimbabwe and elsewhere. However, the SHINE baseline data was collected over three calendar years, from 2012 to 2015. Therefore, all months are represented in our analyses since households were interviewed year-round. In addition to the validity analyses that looked at differences in scores by season (Table 5), we also ran multiple sensitivity analyses of separate factor analyses with groups of households that were interviewed by calendar quarter, by dry season and rainy season, and by hungry season and plenty season. In all instances, the same items were loaded on the same dimensions as our main analyses. Furthermore, we had another sample of the households at a different time point. As shown in Table 6, when we ran factor analysis at the 18-month mark, the results we report in this paper remained consistent in terms of dimensions and item variables. All these analyses greatly strengthen our confidence in the structural validity of our measures and the dimensions identified for FI and WI.

## 5. Conclusions

We developed new and culturally-specific measures for each FI and WI among rural Zimbabwean households. In accordance with the theoretical definitions of FI and WI, each measure was multidimensional, with three distinct dimensions retained. FI was characterized by ‘poor food access’, ‘household shocks’ and ‘low food quality and availability’. WI was characterized by ‘poor water access’, ‘poor water quality’, and ‘low water reliability’. The application of such multidimensional measures will make it possible to pinpoint the components of FI and WI that impact health, and may facilitate the provision of better interventions to households in need of specific food and water support. These measures will also contribute to transformative WASH actions, and transdisciplinary FI and WI mitigation efforts.

## Figures and Tables

**Figure 1 ijerph-18-06020-f001:**
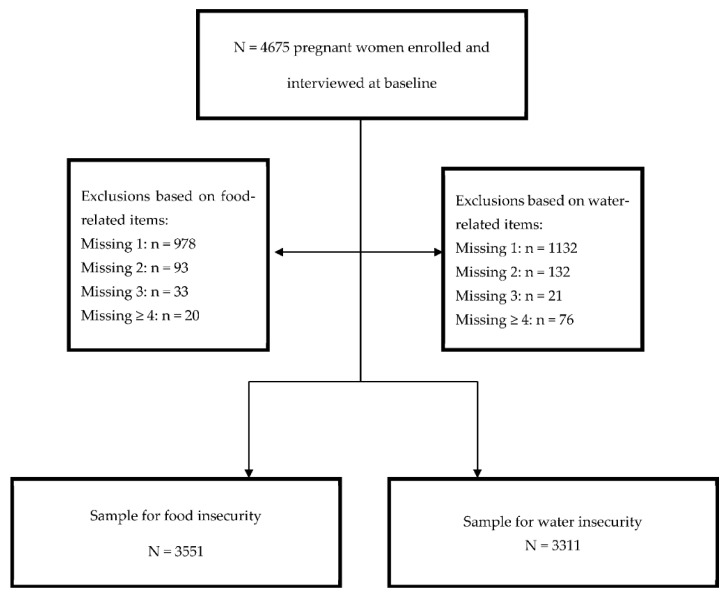
Sample selection for FI and WI factor analyses.

**Figure 2 ijerph-18-06020-f002:**
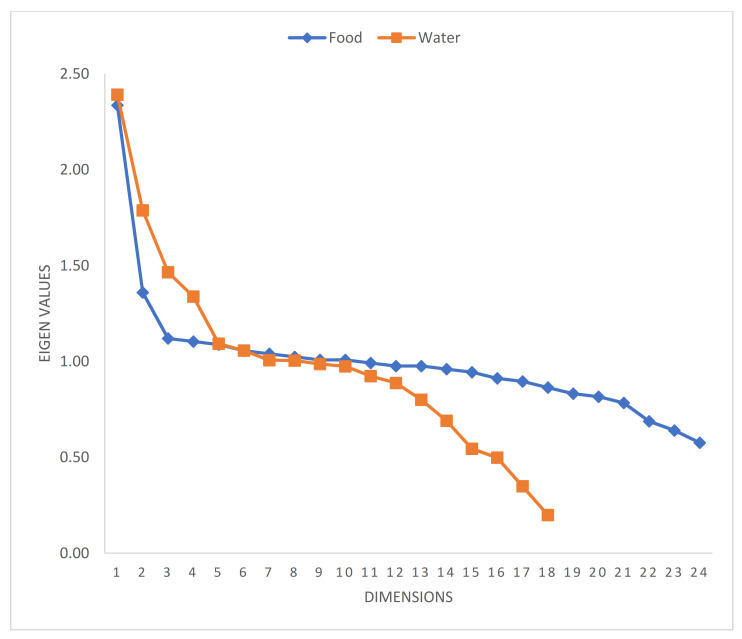
Scree plots of food and water dimensions.

**Table 1 ijerph-18-06020-t001:** Complete set of item variables from the Sanitation Hygiene and Infant Nutrition Efficacy (SHINE) trial considered for each dimension of household food insecurity and water insecurity, collected at baseline from November 2012 to March 2015.

Insecurity	Dimension	Item Variable *	Description of Item Variable	Data Collection	Variable Parameterization **	Recall Period
**Food**	**Availability**	Stock of staple food	Number of days of staple foods available for all household members.	Reported by participating women	8 ordered categories (days): (1) 0–7, (2) 8–30, (3) 31–60, (4) 61–90, (5) 91–120, (6) 121–180, (7) 181–270, (8) >270	Present
		Garden	Household has a garden to grow fruits and/or vegetables	Reported by participating women	Yes vs. No	Present
		Leftover food ^a^	Food left over from last cooking time for the household.	Observed by interviewer	Yes vs. No	Present
	**Access**	Food not preferred	Household relies on inexpensive and less prefered food.	Reported by participating women	Yes vs. No	Month
		Insufficient food	Household members skip entire days without eating, limit portion sizes or reduce number of meals.	Reported by participating women	Yes vs. No	Month
		Food help	Household requires help from family and friends for food, sends members to eat elsewhere or begs for food.	Reported by participating women	Yes vs. No	Month
		Food on credit	Household borrows or purchases food on credit.	Reported by participating women	Yes vs. No	Month
		Assets sold for food	Household sells possessions and assets to afford food.	Reported by participating women	Yes vs. No	Past 3–4 months
		Time to food market	Time taken to go from the homestead to the closest food market (one-way).	Estimated by participating women	4 ordered categories (minutes): (1) 0–20, (2) 21–40, (3) 41–60, (4) >60	Usual
		Transportation ^a^	Method of transportation to get to the food market.	Reported by participating women	Categorical: On foot vs. Other (Motor vehicle or bicycle)	Usual
	**Utilization**	Household diet diversity	Household dietary diversity score calculated from FFQ with reported consumption of 12 food groups.	Reported by participating women	<6 food groups vs. ≥6 food groups	Past 24 h
		Handwashing	Responded volunteered answer “Washed hands prior to food handling” when asked reasons for handwashing.	Reported by participating women	Yes vs. No	Usual
		Food storage location	Location of leftover food: on the floor or in an elevated position.	Observed by interviewer	Elevated vs. floor	Present
		Food container ^a^	Leftover foods stored in open or closed containers.	Observed by interviewer	Covered vs. Not covered	Present
	**Reliability**	Social shocks	Household experienced conflict, legal issues or divorce.	Reported by participating women	Yes vs. No	Past year
		Economic shocks	Household experienced job loss, business failure or loss of assets.	Reported by participating women	Yes vs. No	Past year
		Agriculture shocks	Household experienced loss of crops and/or livestock.	Reported by participating women	Yes vs. No	Past year
		Health shocks	Household members experienced death, disease and/or injury.	Reported by participating women	Yes vs. No	Past year
**Water**	**Availability**	Water volume	Total quantity of water immediately available to household.	Calculated from observed data	4 categories: (1) 0–20 L, (2) >20–40 L, (3) >40–60 L, (4) >60 L	Present
		Irrigation water ^a^	Household has access to water for irrigation.	Reported by participating women	Yes vs. No	Present
	**Access**	Time to drinking source	Time taken to get to main source of drinking water from homestead (one-way).	Estimated by participating women	<15 min vs. ≥15 min	Usual
		Distance to drinking source	Distance to main drinking water source from homestead (one-way).	Estimated by participating women	<1000 m vs. ≥1000 m	Usual
		Time to non-drinking source	Time taken to get to main source of non-drinking water (one-way).	Estimated by participating women	<15 min vs. ≥15 min	Usual
		Distance to non-drinking source	Distance to main source of water for non-drinking purposes(one-way).	Estimated by participating women	<1000 m vs. ≥1000 m	Usual
		Water purchase ^a^	Does the household purchase water usually?	Reported by participating women	Yes vs. No	Usual
	**Utilization**	Drinking source type	Water source: (1) piped into dwelling, (2) piped into yard or plot, (3) piped into public tap or standpipe, (4) borehole, (5) protected deep well, (6) unprotected deep well, (7) protected shallow well, (8) unprotected shallow well, (9) improvised shallow well, (10) protected spring, (11) unprotected spring, (12) surface water from river/dam/stream/lake, (13) river bank, (14) rainwater harvester, (15) water truck/Bowser, (16) bottled	Reported by participating women	3 categories: (1) piped, protected sources, bottled, water truck and rainwater harvester, (2) unprotected ground water and improvised water sources, (3) surface water and river banks.	Usual
		Non-drinking source type	Same as drinking source type above	Same as above	Same as above	Usual
		Water satisfaction	Satisfaction with smell, color and taste of water from main water sources.	Reported by participating women	3 categories: Satisfied/Neutral/Unsatisfied	Usual
		Water treatment	Household treats water to make it safe for consumption e.g., boiling, bleaching, use chlorine, etc.	Reported by participating women	Yes vs. No	Usual
		Water container	Water meant for drinking was kept in covered containers.	Observed by interviewer	Covered vs. Not covered	Present
	**Reliability**	Drinking water frequency	Frequency at which main source for drinking water runs dry.	Reported by participating women	Ever vs. Never	Anytime over past year
		Non-drinking water frequency	Frequency at which main source for non-drinking water runs dry.	Reported by participating women	Ever vs. Never	Anytime over past year

* All item variables were either dichotomous or ordered categorical, and reverse coded so that insecurity scored higher; ** Parameterization of variables as used in the subsequent quantitative analyses in this study; ^a^ Variables excluded in the subsequent steps of factor analysis if categories were too small (≤5%) or too common (≥95%).

**Table 2 ijerph-18-06020-t002:** Validity assessments for dimension scores of food insecurity and water insecurity.

Type of Validity	Purpose	Assessment Methods
**Internal**	To determine the extent to which the dimensions obtained are consistent within the sample and across time.	1. CFA with a sub-sample of the population at baseline to cross-validate the number of dimensions and loading patterns.
		2. CFA with the same households reporting information at 18 months to cross-validate the results across time.
**Predictive**	To determine the extent to which the dimension scores predict known related outcomes.	Linear regression to estimate the associations between dimensions of:
		2. Water insecurity and perceived health status and depression symptomatology.
		1. Food insecurity and perceived health status and depression symptomatology.
**Discriminant**	To determine the extent to which the dimension scores are differentiated as expected according to known groups.	Tests of differentiation between dimension scores across known groups using linear regression:
		1. Food insecurity: season (hungry vs plenty), SES-status, HIV-status.
		2. Water insecurity: season (rainy vs dry), wealth index, HIV-status.
**Convergent**	To determine the extent to which the dimension scores are associated with other constructs that are closely related.	Linear regression to estimate association between dimensions of:
		1. Food insecurity and receiving food aid.
		2. Water insecurity and frequency of water collection.

**Table 3 ijerph-18-06020-t003:** Socio-demographic characteristics of households included in analyses.

Characteristics	Food Sample	Water Sample
**Socio-demographic *, *N***	**3551**	**3311**
Trial arm, *n* (%)		
SOC	803 (22.61)	772 (22.61)
IYCF	872 (24.56)	780 (24.56)
WASH	914 (25.74)	856 (25.74)
WASH + IYCF	962 (27.09)	903 (27.09)
Living with HIV, *n/N* (%)	556/3537 (15.66)	524/3297 (15.66)
Rainy season at interview, *n/N* (%)	-	1473/3310 (45.17)
Hungry season at interview, *n/N*(%)	1030/3545 (29.01)	942 (29.01)
Woman’s education, *n* (%)		
Primary	640 (18.02)	596 (18.02)
Some secondary	1263 (35.57)	1199 (35.57)
Completed secondary	1510 (42.52)	1382 (42.52)
Women employed, *n*/*N*(%)	298/3241 (8.39)	284/3302 (8.39)
SES tercile, *n* (%)		
lower	1141 (32.13)	1079 (32.13)
middle	1192 (33.57)	1117 (33.57)
upper	1214 (34.19)	1111 (34.19)
Women partnered, *n/N*(%)	3233/3384 (91.04)	3015/3156 (91.04)
Parous ^1^, *n*/*N*(%)	1885/2254 (53.08)	1964/2330 (59.32)
Religion: Apostolic, *n*/*N*(%)	1596/3411 (44.95)	1513/3176 (44.95)
Depression, *n*/*N*(%)	201/3485 (5.66)	155/3170 (5.66)
Woman’s age, *n*/mean (SD)	3401/26.42 (6.72)	3171/26.32 (6.68)
Household size, *n*/median (IQR)	3432/5 (3)	3199/5 (3)
Perceived health status ^1^, *n*/mean (SD)	3065/3.42 (0.99)	2874/3.42 (0.99)
**Food item variables **, *N***	**3551**	
Staple food stocks		
>270 days	402 (11.32)	-
181–270 days	370 (10.42)	-
121–180 days	584 (16.45)	-
91–120 days	296 (8.34)	-
61–90 days	447 (12.59)	-
31–60 days	459 (12.93)	-
8–30 days	562 (15.83)	-
0–7 days	431 (12.14)	-
Garden, *N* (%)	2904 (81.78)	-
Leftover food, *N* (%)	1705 (48.01)	-
Food not preferred	2478 (69.78)	-
Insufficient food	866 (24.39)	-
Food help	477 (13.43)	-
Food on credit	898 (25.29)	-
Assets sold for food	404 (11.38)	-
Time to food market, *n* (%)		
0–20 min	985 (27.74)	-
21–40 min	842 (23.71)	-
41–60 min	1005 (28.3)	-
>1 h	719 (20.25)	-
Transportation, *n* (%)		
Bicycle/Motor	150 (4.22)	-
Walking	3384 (95.3)	-
Household meets diet diversity	2581 (72.68)	-
Handwashing prior to food handling	3198 (90.06)	-
Food storage location		
Elevated position	2091 (58.88)	-
On floor	1460 (41.12)	-
Food container covered	3513 (98.93)	-
Social shocks	261 (7.35)	-
Economic shocks	472 (13.29)	-
Agriculture shocks	893 (25.15)	-
Health shocks	1670 (47.03)	-
**Water item variables **, *N***		**3311**
Volume		
>60 L	-	370 (11.17)
41 to 60 L	-	458 (13.83)
21 to 40 L	-	1093 (33.01)
0–20 L	-	1390 (41.98)
Water for irrigation	-	699 (21.11)
One-way ≤ 15 min to drinking water source	-	2411 (72.82)
Distance ≤ 1000 m to drinking source	-	2902 (87.65)
One-way ≤ 15 min to non-drinking source	-	2470 (74.6)
Distance ≤ 1000 m to non-drinking source	-	2918 (88.13)
Purchase of water	-	5 (0.15)
Type of drinking source		
Improved (piped, protected)	-	2097 (63.33)
Unprotected ground	-	954 (28.81)
Surface water	-	260 (7.85)
Type of non-drinking source		
Improved (piped, protected)	-	1105 (33.37)
Unprotected ground	-	970 (29.3)
Surface water	-	1236 (37.33)
Satisfaction with main water source		
Satisfied	-	2677 (80.85)
Neither satisfied not unsatisfied	-	343 (10.36)
Unsatisfied	-	291 (8.79)
Drinking water treated	-	412 (12.44)
Drinking water containers covered	-	2143 (64.72)
Drinking water always available	-	2892 (87.35)
Water for non-drinking purposes always available	-	2951 (89.13)

* All socio-demographic variables had <5% missing data unless otherwise stated. ** All variables represent complete data as n(%) unless otherwise stated. ^1^ Missing data >5%. - Empty cells imply that the variables were not described for that sample.

**Table 4 ijerph-18-06020-t004:** Squared correlation ratios, eigenvalues, percentage variances and descriptive statistics of multiple correspondence analysis for final dimensions in each food insecurity and water insecurity measures.

Food Insecurity (*N* = 3551)				Water Insecurity (*N* = 3311)			
MCA Food Dimensions	1	2	3	MCA Water Dimensions	1	2	3
Dimension Names	Poor Food Access	Household Shocks	Low Food Availability and Quality	Dimension Names	Poor Water Access	Poor Water Quality	Low Water Reliability
**Stock of staple food**	0.11	0.05	**0.31**	Water volume	0.01	0	0.01
**Garden**	0	0	**0.32**	**Time to drinking source**	**0.56**	0	0
**Food not preferred**	**0.48**	0.01	0.0	**Distance to drinking source**	**0.48**	0	0
**Insufficient food**	**0.52**	0	0.01	**Time to non-drinking source**	**0.58**	0	0
**Food help**	**0.47**	0	0	**Distance to non-drinking source**	**0.49**	0	0
**Food on credit**	**0.47**	0.01	0	**Drinking source**	0.09	**0.76**	0.06
Assets sold for food	0.02	0.04	0.05	**Non-drinking source**	0.10	**0.66**	0.01
Time to food market	0.03	0.05	0.11	**Water satisfaction**	0.04	**0.30**	0
Handwashing	0	0.01	0.02	Water treatment	0	0.04	0
Food storage location	0.01	0	0.11	Water container	0	0	0.01
**Household diet diversity**	0.03	0.06	**0.25**	**Frequency of availability of drinking water at the source**	0	0	**0.70**
Social shocks	0.01	0.07	0.03				
**Economic shocks**	0.02	**0.32**	0.01	**Frequency of availability of non-drinking water at the source**	0	0	**0.70**
**Agriculture shocks**	0.01	**0.36**	0				
**Health shocks**	0	**0.34**	0				
**Eigenvalue**	2.17	1.32	1.22	**Eigenvalue**	2.36	1.78	1.50
**% variance**	9.06	5.93	5.13	**% variance**	13.29	9.93	8.14
**Median score (IQR)***	−0.32 (1.35)	−0.14 (1.36)	−0.07 (1.34)	**Median (IQR)**	−0.48 (1.40)	−0.28 (2.01)	−0.32 (0.26)
**Min, max score ***	−1.33, 3.39	−2.25, 3.41	−2.81, 3.93	**Min, max**	−1.04, 3.63	−1.41, 2.43	−1.46, 3.46

Items in bold are retained as relevant (squared correlation ratio ≥ 0.2) and further indicate which dimension they load on. * Scores are standard dimension scores obtained from post-estimation commands. Higher positive scores on dimensions are indicative of higher insecurity.

**Table 5 ijerph-18-06020-t005:** Internal validity of multidimensional food insecurity and water insecurity measures.

Measure	Food Insecurity		Water Insecurity	
Groups	Baseline Test Sample ^1^	18 months ^2^	Baseline Test Sample ^1^	18 months
N	2132	3612	1998	3879
RMSEA ≤ 0.05 (range: 0, 0.10)	**0.04** **(0.03, 0.04)**	**0.04** **(0.03, 0.04)**	**0.04** **(0.04, 0.05)**	0.06 (0.06, 0.07)
CFI ≥ 0.95	0.88	0.90	**0.96**	**0.97**
TLI ≥ 0.90	0.87	0.88	**0.95**	**0.97**
SRMR ≤ 0.08	**0.07**	**0.08**	**0.07**	0.09

^1^ Baseline test sample refers to a sub-sample of the complete case used to confirm the measures. We used 40% of the initial baseline sample as a training dataset and ran exploratory factor analysis; the remaining mutually exclusive 60% of the sample was then used as the testing dataset to confirm the exploratory findings via confirmatory factor analysis. ^2^ The 18-month time point refers to 18 months after the pregnant woman gave birth. Therefore, this sample could be between 19–24 months post baseline interview. Bolded values represent satisfactory model fit statistic. RMSEA= root mean square error of approximation; CFI = comparative fit index; TLI = Tucker Lewis index; SRMR = Standardized Root Mean Square Residual.

**Table 6 ijerph-18-06020-t006:** Predictive, discriminant and convergent validity of food insecurity and water insecurity measures.

			Food Insecurity (N = 3551) β [95% CI]				Water Insecurity (N = 3311) β [95% CI]			
**Validity**	Variable	Expected direction	*n/N* Median (IQR)	**Poor food** **access**	**Household shocks**	**Low food quality and availability**	*n/N* Median (IQR)	Poor water access	Poor water quality	Low water reliability
				−0.38 (1.34)	−0.18 (1.35)	−0.08 (1.40)		−0.47 (1.40)	−0.27 (1.99)	−0.33 (0.28)
**Predictive**	^a^ Perceived health status	Negative	3065	**−0.16**	**−0.13**	**−0.05**	2874	**−0.04**	**−0.04**	**−0.07**
				**[−0.19, −0.12]**	**[−0.17, −0.09]**	**[−0.08, −0.01]**		**[−0.08, −0.01]**	**[−0.08, −0.01]**	**[−0.11, −0.03]**
	^a^ Depression	Positive	3485	**0.52**	**0.37**	**0.26**	3236	**0.22**	**0.22**	**0.23**
				**[0.38, 0.66]**	**[0.23, 0.51]**	**[0.12, 0.41]**		**[0.07, 0.36]**	**[0.07, 0.36]**	**[0.08, 0.37]**
**Discriminant**	^b^ Season: Hungry vs. Plenty	Positive	1030/3545	**0.12**	0.04	**0.25**	-	-	-	-
				**[0.05, 0.20]**	[−0.03, 0.12]	**[0.18, 0.32]**				
	^b^ Season: Dry vs. Rainy	Positive	-	**-**	-	**-**	1837/3310	**0.16**	0.03	−0.11
								**[0.09, 0.22]**	[−0.04, 0.10]	[−0.18, −0.04]
	^c^ Middle SES vs. High SES	Positive	1192/3547	**0.29**	−0.14	**0.27**	1111/3307	0.05	**0.18**	0.05
				**[0.21, 0.37]**	[−0.22, −0.06]	**[0.19, 0.35]**		[−0.03, 0.13]	**[0.09, 0.26]**	[−0.03, 0.14]
	^c^ Low SES vs. High SES	Positive	1141/3548	**0.56**	−0.27	**0.61**	1117/3308	**0.15**	**0.32**	0.01
				**[0.48, 0.64]**	[−0.35, −0.19]	**[0.53, 0.69]**		**[0.07, 0.24]**	**[0.24, 0.40]**	[−0.07, 0.10]
	^d^ HIV-positive vs. HIV negative	Positive	556/3547	**0.21**	−0.01	**0.16**	524/3297	0.06	0.05	−0.02
				**[0.12, 0.30]**	[−0.05, 0.02]	**[0.07, 0.24]**		[−0.03, 0.15]	[−0.04, 0.15]	[−0.11, 0.07]
**Convergent**	^c^ Receive food aid vs. no food aid	Positive	316/3551	0.12	**0.19**	0.06	-	-	-	-
				[0.00, 0.23]	**[0.07, 0.30]**	[−0.05, 0.18]				
	^a^ Water collection- Weekly vs. Daily	Positive	-	-	-	-	488/3306	**0.17**	0	0.05
								**[0.07, 0.27]**	[−0.09, 0.10]	[−0.05, 0.15]
	^a^ Water collection- Monthly vs. Daily	Positive	-	-	-	-	140/3307	**0.24**	−0.09	−0.08
								**[0.07, 0.41]**	[−0.26, 0.08]	[−0.25, 0.09]

Values in bold represent statistically significant (at *p* < 0.05) associations in the expected directions from linear regressions. ^a^ Information self-reported by mothers for their status and activity; ^b^ Based on date of interview recorded by data collector; ^c^ Household-level information; ^d^ Mother’s information obtained from rapid blood test.

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
