# Peer review of "Food Insecurity and Water Insecurity in Rural Zimbabwe: Development of Multidimensional Household Measures"

_ijerph, 2021, doi:10.3390/ijerph18116020_

Round 1

Reviewer 1 Report

Food Insecurity and Water Insecurity in Rural Zimbabwe: Development of Multidimensional Household Measures

Summary: This is an ambitious paper that adds depth to a vibrant academic conversation centered on improving measures of household food and water insecurity. By developing and validating separate multidimensional measures for household FI and WI in the context of rural Zimbabwe, this manuscript pushes this body of research forward.

Broad strengths and weaknesses: The scope of this paper is at once a strength and a weakness. I appreciated that the authors make it clear that the point of exploring multidimensional aspects of FI and WI is because of the utility for sanitation/nutrition interventions.

However, within the abstract and introduction, such a heavy focus on undernutrition is somewhat misleading. Yes, this is the background, but I think the real meat of the paper is about assessing the relevant dimensions of WI and FI.

The authors do a nice job summarizing the state of FI/WI metrics in the introduction and this seems like a more appropriate initial background providing the rationale for this type of work. I think it would be beneficial to then highlight that this manuscript specifically examines the internal, convergent, and discriminant validity of FI and WI metrics and then round off with the broader picture of why this work is important for better evaluating interventions - so just a little bit of re-organization of the same material might make it a smoother/more intuitive read in the beginning of the paper

The main text in the manuscript is dense but informative and I think the "Discussion" adequately captures both the strengths and weaknesses of the analysis/results. My one question is whether the authors can estimate the time commitment for all the suggested components of the FI and WI metrics?

HWISE recently assessed the feasibility of integrating WI questions into larger surveys and there conclusions may be relevant (https://www.ajtmh.org/view/journals/tpmd/104/1/article-p391.xml#:~:text=The%204%2Ditem%20HWISE%2D4,with%20the%20full%20HWISE%20Scale.)

Specific changes:

Ref not found on pg 4 line 156 and then a number of times afterwards.

The strikeout of the variables is a bit odd. Might be better to just specify that those variables were excluded in the text.

Section 2.5.3 - create a new paragraph beginning with "For predictive, discriminate, and convergent validity…"

Opening line in the "Discussion" is repetitive:

The goal of this study was to develop new measures of FI and WI that are cognizant of their multidimensionality to advance the discussion on impactful nutrition and health interventions for vulnerable populations by considering the multidimensionality of FI and WI.

Author Response

Thank you for this review. It is great to see that our research is considered of value to the current state of knowledge. We appreciate your time and effort on a thorough review of our paper. We have addressed all of the questions and  incorporated all changes suggested as described in the attached word document. We believe these modifications have improved our manuscript. Please see the attachment.

Reviewer 2 Report

“Food Insecurity and Water Insecurity in Rural Zimbabwe: Development of Multidimensional Household Measures” submitted to IJERPH (IJERPH 1207832)

The paper uses household-level data collected as part of a randomized control trial to construct and validate measures of food insecurity (FI) and water insecurity (WI) in a population exhibiting high undernutrition in Zimbabwe. It utilizes rich and comprehensive survey data (for example I believe more comprehensive than a typical DHS for these areas) and reflects a well-designed study that sought to characterize FI and WI. Using factor analysis the authors develop a set of measures for each insecurity finding they are not well represented by single factors. The indicators will likely be useful for characterizing relationships between FI and WI with child undernutrition (which seems to be part of the main study) in future work and, possibly, for others working in this area – at least as a methodological approach if not a set of indicators to mimic exactly.

Main comments

  1. It appears FI is better studied elsewhere in the literature than WI. For example: “Decades of work on FI have produced a diverse range of metrics at the household level.” As such, it seems the case for why developing another set of measures will add value is particularly important to make in justifying this paper. Related, on line 78 indicated FAO recommends development and application of several indicators – does this mean they recommend each study developing it’s own indicators (as you do here)?
  2. Is the objective of the paper for others to use this indicator? This process? Pay more attention to x, y, z when constructing similar measures. More clarity on that would have been helpful, I believe. (4. Discussion says something about the objective but it was not clear to me what that meant).
  3. Line 46 – mentioned that FI and WI often coexist, would it not be useful to characterize that within the sample studied here, a clear benefit of your set-up? How are the elements of FI and WI correlated with one another? I believe some comparison of the measures would be a valuable addition to this literature.

  1. 5.3. Different perspectives exist as to whether the variables you select for validation are “exogenous”. Indeed you indicate later that there may be bi-directional influences between HIV (possibly operating through reduced labor effort etc). The same is logically the case for poor self-reported health or depression. I’m not suggesting these are not informative correlations to look at, just that it may be more appropriate to acknowledge these possibilities in interpretation particularly when discussing causal impacts.
  2. I think Figure 2 scree plot suggests more than three dimensions are relevant for water; looks like five.
  3. Some more thinking about the role and potential problems of seasonality (of the two types indicated) is warranted. Is it valid to construct this sort of measure when households are in different seasons? Many surveys do this (eg DHS) so it may be okay but still warrants careful consideration (and maybe there are some lessons here for DHS type surveys).
  4. I was unable to easily follow the samples used for internal validity. How as the test subsample selected (presumably some sort of stratified random?). Regarding the 18-month sample, is this 18 months after baseline? If so, that presents an interesting opportunity since those interviews are, by definition, nearly all in opposite seasons (hungry/not, wet/dry) than the baseline for the same household (see prior point). If the 18-month refers to the kid and wasn’t at 18 months then it means they would be at a mix of different seasons relative to baseline so that would need to be considered in interpretation.
  5. 3: More information is needed for the reader to judge the selectivity of the final sample ~3500 compared to baseline samples of 4500+, more than 20% loss. Later in the paper some sensitivity is carried out for larger samples, so it does not appear to be a problem of large bias in the results but this needs to be made clear. How is the sample different.
  6. There are many references that did not resolve in the pdf I was provided. Line numbers also did not come through.

Other comments

  1. Abstract line 23: “undernutrition is causally related to resources including food insecurity (FI) and water insecurity (WI)” – are these concepts commonly referred to as “resources”
  2. 5.2 – I did not understand the concept of “negative variance”, but am not an expert in these statistics. It may be useful to indicate why/how this is possible.
  3. Line 102 – 30% is BOTH undernourished AND obtain water from unprotected services (since there are two cites it was not clear to me these populations overlap so that 30% have both conditions).
  4. Line 125: Households consisting of women and children – so there are no male partners in the households in this study?
  5. In Table 1 indicate reasons for exclusion of the different variables in strikeout (eg <5% or > 95% as described in the text).
  6. Table 3 – I did not understand what “food not preferred” meant (but may have missed the full definition elsewhere).
  7. Need to make clearer the validity for health indicators is with respect to the condition of the woman/mother (not child, not others in the household, etc), assuming I correctly understood the structure of the questionnaire.
  8. The cumulative variance explained seems low even after including 3 factors – comment on this or possibly compare with other studies. For example, for WI I understand here are 15 factors (Table 4), 3 explain 20% so the remaining 12 explain 80%.
  9. It was not entirely clear to me why this being a rigorous RCT was a strength for this paper since it primarily uses baseline data.
  10. 3 – the indicators used to reflect differences in “intrahousehold allocation of food” were not intuitive and need further justification/clarification.
  11. 3 I wondered whether diarrheal episodes might have been a useful validation variable to get at the validity of the water quality dimension.
  12. In some ways the first strength mentioned is also the first weakness mentioned, somewhat contradictory. Related, in conclusion, refrain from calling it a “representative” sample since that was already one of the weaknesses you highlighted.

Author Response

We thank the reviewer for their time and expertise on giving us such a thorough review. We have addressed all of the questions and  incorporated all changes suggested as described in blue below. We strongly feel that this review has improved our manuscript. Please see the attachment for details on changes made.

Reviewer 3 Report

The authors show a rigorous analysis and the limitations of the study are properly described. This manuscript might be relevant and important for the audience to read and to take into consideration in the future. I would accept the manuscript after minor revision:

  • Tables 1 and 2 in supplementary figures should be named Table S1 and S2.
  • All the tables and figures should be cited throughout the text.
  • Most of the references are missing in the text. Instead, they are shown as (Error! reference source not found)
  • In the manuscript it is written: “Supplementary Materials: The following are available online at www.mdpi.com/xxx/s1, Figure S1; Table S1; Table S2”. That does not match with the Supplementary material which are 3 tables.

Author Response

We would like to thank reviewer 3 for their time and expertise in reviewing our manuscript. We have addressed all of the questions and incorporated all changes suggested as described in the word document attached. We believe that the changes we made based on the comments received have made our manuscript better. Please see the attachment for details.

Round 2

Reviewer 2 Report

The revisions and response were thoughtfully done, clarifying my misunderstandings and the overall message of the paper. Some minor items.

  • Is the cutoff 4/96 or 5/95, table and text do not agree on this (my hunch is in text you meant to put greater or equal to?)
  • We still disagree on variance - as a statistical measure variance cannot be negative; perhaps you mean covariance? 
  • references still did not resolve correctly in the pdf i was provided so you should check final proof prepared by the journal

Author Response

Thank you so much for giving us the opportunity to further improve our paper. We have addressed the remaining concerns. Please see the attachment.
